# Peer review of "Bayesian Analysis of Population Health Data"

_mathematics, doi:10.3390/math9050577_

Round 1
Reviewer 1 Report
Review on ‘Bayesian analysis of population health data’
This paper focus on applications of Bayesian hierarchical models to analyze ischemic stroke incidence. The problem may have practical interest. However, the paper does not allow one to evaluate the real novelty of the results obtained in it:
- The authors seem to be completely unaware about the vast amount of work that has been done about the mixed effect parameters logistic regression model. The introduction is a short, maybe the author could describe the state of art with more details. All theoretical results are in the frame of the well-known regression models.
- Weakness of main contributions. The authors in abstract and introduction should explain better key advances and novelty in new developed model.
- This is a commonly formalized problem which is implemented in different computer softwares. I suggest the authors to include references on computer simulations, and make paper more interesting for the reader.
Specific comments:
- What it is ‘the regional variation’?
- Line 91 Eq. (1) unclear, pi isn’t defined.
- (2) presents the logistic regression with random intercept (only!). Generally speaking, there is usually not much information available on the random effects, beyond a random intercept, when the number of repeated measurements is relatively small.
- (3) in previous page is declared that ‘considers random effects as conditionally independent and identically distributed random variables - ÉŁi’.
- Lines bellow 94 till 102 are declarative and not related with this study.
- Line 137, D isn’t determined.
- Section 2.3 presents well-known information (could be valuable without imposing any distributional assumptions on the random effects).
Author Response
Please, find your reply to reviewer 1 in the file attached.

Reviewer 2 Report
This paper addresses a very important topic of investigation, namely a Bayesian analysis of population health data. The Bayesian approach essentially uses the observed distribution to show the underlying distribution's parameters, and is useful in areas such as statistics, finance, public health, etc. The manuscript proposes an innovative technique, which can capture key aspects of population health data in Poland. The posterior summaries for the parameters and hyper-parameters of the models look great in Table 2.
However, the authors need to address the following points:
- What I see is that the authors have used unbalanced data. For example, lines 186 and 225 show that the percentage of ischemic stroke is very low. An unbalanced sample can lead to unrealistic conclusions. Is the author concerned about the imbalanced data issue in this case? If so, how are the authors addressing it? The authors should clarify how to deal with unbalanced data here.
- It would be great if the authors find out the important and unimportant variable for causing ischemic stroke from Table 2.
- I would recommend authors to add a conclusion that describes the importance of their work.
Author Response
Please, find our reply to reviewer 2 in the file attached.

Round 2
Reviewer 1 Report
Thank. All the points raised have been addressed satisfactorily. The study is relevant to the mission of the journal and worthy of consideration for publication in Mathematics.